# Insights from circulating microRNAs in cardiovascular entities in turner syndrome patients

**Masood Abu-Halima**[1]*, **Felix Sebastian Oberhoffer**[2], **Mohammed Abd El Rahman**[2], **Anna-Maria Jung**[3], **Michael Zemlin**[3], **Tilman R. Rohrer**[3], **Mustafa Kahraman**[4], **Andreas Keller**[4], **Eckart Meese**[1☉], **Hashim Abdul-Khaliq**[2☉]

1 Institute of Human Genetics, Saarland University, Homburg/Saar, Germany, 2 Department of Pediatric Cardiology, Saarland University Medical Center, Homburg/Saar, Germany, 3 Department of Pediatric Endocrinology, Saarland University Medical Center, Homburg/Saar, Germany, 4 Chair for Clinical Bioinformatics, Saarland University, Saarbruecken, Germany

☉ These authors contributed equally to this work.
* masood@daad-alumni.de

**Data Availability Statement:** All relevant data are within the paper and its Supporting Information files.

## Abstract

### Background

Turner syndrome (TS) is a chromosomal disorder, in which a female is partially or entirely missing one of the two X chromosomes, with a prevalence of 1:2500 live female births. The present study aims to identify a circulating microRNA (miRNA) signature for TS patients with and without congenital heart disease (CHD).

### Methods

Microarray platform interrogating 2549 miRNAs were used to detect the miRNA abundance levels in the blood of 33 TS patients and 14 age-matched healthy volunteer controls (HVs). The differentially abundant miRNAs between the two groups were further validated by RT-qPCR.

### Results

We identified 60 differentially abundant miRNA in the blood of TS patients compared to HVs, from which, 41 and 19 miRNAs showed a higher and a lower abundance levels in TS patients compared to HVs, respectively. RT-qPCR confirmed the significantly higher abundance levels of eight miRNAs namely miR-374b-5p, miR-199a-5p, miR-340-3p, miR-125b-5p, miR-30e-3p, miR-126-3p, miR-5695, and miR-26b-5p in TS patients as compared with the HVs. The abundance level of miR-5695 was higher in TS patients displaying CHD as compared to TS patients without CHD (p = 0.0265; log2-fold change 1.99); whereas, the abundance level of miR-126-3p was lower in TS patients with congenital aortic valve disease (AVD) compared to TS patients without BAV (p = 0.0139, log2-fold change 1.52). The clinical feature statistics revealed that miR-126-3p had a significant correlation with sinotubular junction Z-score (r = 0.42; p = 0.0154).

**Funding:** This study was funded by Competence Network for Congenital Heart Defects [Nr. 01GI0601 (2014)] to HA-K, German Centre for Cardiovascular Research (DZHK) [Nr. 81X2800112 (2015)] to H-AK, and Hedwig-Stalter-Stiftung (2016) to MA-H.

**Competing interests:** The authors declare that they have no competing interests.

## Conclusion

The identified circulating miRNAs signature for TS patients with manifestations associated with cardiovascular diseases provide new insights into the molecular mechanism of TS that may guide the development of novel diagnostic approaches.

## Background

Turner syndrome (TS) is a chromosomal disorder, in which a female is partially or entirely missing one of the two X chromosomes, with a prevalence of 1:2500 live female births [1]. Females with TS display an increased cardiovascular risk, and overall mortality is reported to be higher than in the general population [2]. In TS patients, congenital heart diseases (CHD) like coarctation of the aorta (CoA) and bicuspid aortic valve (BAV) are common [3]. The prevalence of diabetes, lipid anomalies, arterial hypertension, and excess weight is elevated in TS patients as well [3, 4]. Recently, widespread epigenetic and gene expression studies in patients with TS have been carried out [5, 6]. These studies indicated that many genes are involved in epigenetic regulatory processes of TS and that many genes displayed a wide range of expression variation, including genes deregulated in TS. This gene expression variation can lead to a wide range of different phenotypes observed among TS patients. The resulting phenotype is caused by a combination of two possible factors: a genomic imbalance due to gene deletions and an additive influence from related genes within the gene network, causing altered gene expression regulation due to a lack of one of the sex chromosomes. To date, TS research has focused primarily on genetic and chromosomal abnormalities, whereas epigenetic and/or gene expression effects have not been widely studied. MicroRNAs (miR, miRNA) are a novel class of small non-coding RNAs ($\sim$ 22 nucleotides in length), which regulate gene expression post-transcriptionally by repression of translation and/or degradation of mRNA [7]. To date, 2300 true human mature miRNAs have been reported [8], and miRNAs are involved in many, if not all, cellular and biological processes investigated so far [9]. The involvement of the miRNAs in the pathology of diabetes mellitus, obesity, and cardiovascular manifestations have been studied extensively [10, 11]. Further studies demonstrated that changes in the function of miRNAs are closely related to multiple forms of CHD [12–15]. A set of 4 miRNAs (miR-130a, miR-122, miR-486, and miR-718) was correlated with BAV and aortic dilation. These 4 miRNAs were identified to play a role in activating the TGF-β1 pathway and vascular remodeling mediated by vascular endothelial growth factor (VEGF) signaling pathway [16]. To the best of our knowledge, a circulating miRNA signature for TS patients, and the differentially abundant miRNA profile in the TS patients with and without CHD has not been studied yet. Therefore, the determination of miRNAs in these patients could lead to more profound insights into the cardiovascular pathophysiology of TS.

## Methods

### Ethical statement

This study was approved by the Ethics Committee of the Ärztekammer des Saarlandes (State Chamber of Physicians of the German federal state of Saarland), Faktoreistraße 4, 66111 Saarbrücken, Germany, on March 23rd, 2018; approval statement No. 07/18. All patients, their parents or legal guardians provided prior written informed consent.

## Subjects

A total of 33 patients with an approved genetic diagnosis of TS and 14 healthy age-matched volunteer controls (HVs) were enrolled in the study. Patients with TS who were examined regularly either at the Department of Pediatric Cardiology or at the Department of Pediatric Endocrinology of Saarland University Hospital were included in this study. The absence of a particular disorder or any heart abnormality in the HVs was verified by physical examination and two-dimensional echocardiography. The mean age of patients and controls was 17.78 ± 7.88 years. In all study participants, 2.5 ml of venous blood was taken and injected into PAXgene™ tubes (Becton–Dickinson). Tubes were then kept for 24 hours at room temperature to lysis cells, followed by a −20°C storage for multiple days and then at −80°C for long-term storage.

## Assessment of karyotype and cardiovascular morbidity in TS patients

TS patients were screened for CHD through conventional echocardiography. The term congenital aortic valve disease (AVD) includes patients presenting with monocuspid or bicuspid aortic valve. Bodyweight [kg] and height [cm] were measured for each patient. Bodyweight classification was then assessed as follows: Study participants under 18 years of age were classified as normal weight, overweight, or obese depending on body mass index [BMI, kg/m$^2$] percentiles established by Kromeyer-Hauschild et al.,[17] (over-weight $\geq$90[th] percentile, obese $\geq$97[th] percentile), while study participants over 18 years of age were classified as normal weight with a BMI <25 kg/m$^2$, over-weight with a BMI $\geq$25 kg/m$^2$ but <30 kg/m$^2$, and obese with a BMI $\geq$30 kg/m$^2$. Body surface area [BSA, m$^2$] was measured after Mosteller's formula [18]. The presence of carbohydrate metabolism disorders or arterial hypertension was evaluated through a retrospective analysis of clinical records.

## Echocardiographic assessment of left ventricular dimensions

A GE Vivid E9 ultrasound system along with a 2.5–3.5 MHz phased array transducer (GE Healthcare, Fairfield, CT, United States) were used to conduct the echocardiography. The left ventricular dimensions were measured in the parasternal long-axis using M-Mode echocardiography. Z-scores of Cardiac Structures were then calculated according to Pettersen et al. [19]. Additionally, the diameter of the ascending aorta (aortic valve, sinuses, sinotubular junction, ascending aorta) was measured in the parasternal long-axis at the time of systole and calculated the corresponding Z-scores [20].

## Preparation of RNA and RNA quality control

Total RNA including miRNAs from blood samples collected into PAXgene tubes was isolated using PAXgene Blood miRNA Kit on the QIAcube™ robot (Qiagen, Hilden, Germany) as previously described [12]. Briefly, samples were first thawed at room temperature (RT) for 16 hours and centrifuged at 3000x g for 10 minutes at RT. The supernatant was discarded and the pellet was completely dissolved in RNase-free water and centrifuged again at 3000 x g for 10 minutes at RT. The supernatant was discarded again and the pellets was dissolved in 350 µl Buffer BM1 and placed on the QIAcube ™ robot. The procedure was completed according to the manufacturer's recommendations. The RNA concentration and purity were confirmed by the spectrophotometric ratio using absorbance measurements at wavelengths of 260 nm and 280 nm on a NanoDrop-2000 (Thermo Scientific, Waltham, Massachusetts, United States). The integrity of the isolated RNA was analyzed on a RNA Nano 6000 chip using an Agilent Bioanalyzer (Agilent Technologies, Santa Clara, California, United States). Genomic DNA

contamination was removed, and conventional PCR with exon spanning primers was carried out to exclude any residual DNAs in the samples as described previously [21]. RNA Integrity Number (RIN) values of the samples were varied between 7.3 and 8.2.

## MiRNAs profiling by microarray

MiRNA profiling analysis was carried out on the isolated miRNA fraction in 33 patients with TS and 14 HVs using SurePrint™ 8X60K Human v21 miRNA platform (Agilent Technologies) containing probes for the detection of 2549 human miRNAs. One-hundred microliter (μL) of the isolated RNA was labeled and subsequently hybridized to the miRNA microarray chip as previously described [22]. Subsequently, data were imported into R statistical environment software v.2.14.2 for analysis.

## Analysis of miRNAs by RT-qPCR

The abundance level of certain miRNAs was determined using *mi*Script PCR System (Qiagen) on the StepOnePlus™ Real-Time PCR System (Applied Biosystems). In the validation step, 12 miRNAs, including 9 up-regulated miRNAs (miR-374b-5p, miR-199a-5p, miR-340-3p, miR-125b-5p, miR-30e-3p, miR-126-3p, miR-99b-5p, miR-5695, and miR-26b-5p), and another 3 down-regulated miRNAs (miR-6085, miR-5739, and miR-3656) were chosen for RT-qPCR validation based on their higher fold change in patients with TS *versus* HVs (P< 0.05, adjusted FDR, ≥ 2-fold change). Complementary DNA (cDNA) was generated from 250ng of the total RNA and was then diluted to have 0.5 ng/μL cDNA for miRNA detection by qPCR as previously described [22].

## Statistical analysis

Array data were first quantile normalized and then the differentially abundant miRNAs between TS patients and HVs were determined using R software (www.R-project.org). The significance abundance level of each miRNA was calculated by applying an unpaired two-tailed t-test for the miRNAs that exhibited a ≥1.5 fold change. The false-discovery rate (FDR) approach was applied to correct the resulting *P*-values. The relative abundance level for each miRNA was calculated using the $2^{-\Delta\Delta CT}$ equation [23]. RNU6B small nuclear RNA (snRNA) was used an endogenous reference control for normalization purpose as previously described [12, 13, 22, 24–27] and because of its minimum abundance variance between the TS patients and HVs as observed by DataAssist™Software (Applied Biosystems). Correlations were computed using Spearman's regression coefficient and the difference between the groups was analyzed using a Mann-Whitney-U test using GraphPad Prism 7.0. Variables were presented as mean/median ± standard deviation (as indicated in each table). To identify the abundance difference in miRNA levels, an unpaired two-tailed t-test was used to test the mean difference of each miRNA between patients and controls. MiRNAs were considered as differentially abundant if they obtained a *P*-value of < 0.05 and a FDR of ≤ 0.05. We screened for the overlap between the miRNAs and validated target genes using MirTargetLink software (Hamberg et al., 2016).

# Results

## Clinical characteristics of TS patients and HVs

A total of 33 patients with TS and 14 HVs were included in the present study. Clinical characteristics of TS patients and controls are displayed in **Table 1.** The TS patient group was significantly different from the HVs group in terms of height (P<0.001), BMI (kg/m$^2$) (P = 0.014),

**Table 1. Clinical characteristics of patients with TS and HVs.**

| Parameters | TS (n = 33) | HVs (n = 14) | *P* value |
|---|---|---|---|
| Age [years] | 17.11 (8.66/44.13) | 17.86 (12.93/43.82) | 0.306 |
| Height [cm] | 147.52 ± 11.59 | 165.64 ± 6.64 | <0.001** |
| Weight [kg] | 53.05 ± 17.88 | 57.08 ± 8.26 | 0.296 |
| BMI [kg/m$^2$] | 23.90 ± 6.03 | 20.75 ± 2.41 | 0.014* |
| BSA (m$^2$) | 1.46 ± 0.28 | 1.62 ±.14 | 0.015* |
| Z-Scores of Cardiac Structures | | | |
| IVSd | 0.77 ± 0.91 | 0.13 ± 1.24 | 0.056 |
| IVSs | 0.47 ± 0.85 | 0.12 ± 0.74 | 0.180 |
| LVIDd | -0.67 ± 0.92 | -0.33 ± 0.68 | 0.219 |
| LVIDs | -0.43 ± 0.92 | -0.14 ± 0.78 | 0.318 |
| LVPWd | 0.72 (-2.19 / 2.46) | 1.02 (-1.88 / 2.58) | 0.493 |
| LVPWs | -0.50 ± 0.95 | -0.13 ± 0.84 | 0.213 |
| Z-Scores of the Aorta | | | |
| Aortic Valve | 0.31 ± 1.45 | -0.07 ± 0.99 | 0.377 |
| Sinuses | 0.74 ± 1.28 | 0.20 ± 1.23 | 0.197 |
| Sinotubular Junction | 1.08 ± 1.47 | 0.63 ± 1.08 | 0.312 |
| Ascending Aorta | 1.26 ± 1.79 | 0.42 ± 1.53 | 0.132 |
| EDV (BSA) [ml/m$^2$] | 55.25 ± 12.16 | 57.19 ± 9.35 | 0.597 |
| ESV (BSA) [ml/m$^2$] | 18.55 ± 5.60 | 19.08 ± 4.57 | 0.756 |
| SV (BSA) [ml/m$^2$] | 37.52 ± 8.10 | 38.29 ± 6.09 | 0.753 |
| EF [%] | 67.00 ± 6.48 | 67.00 ± 4.76 | 1.000 |
| FS [%] | 37.03 ± 5.23 | 37.07 ± 3.45 | 0.975 |
| LV Mass (BSA) [g/m$^2$] | 71.21 ± 14.57 | 70.41 ± 17.70 | 0.872 |
| MAPSE [mm] | 14.97 ± 2.63 | 15.93 ± 1.77 | 0.219 |
| SBP [mmHg] | 122.58 ± 14.49 | 116.79 ± 9.51 | 0.177 |
| DBP [mmHg] | 75.24 ± 12.62 | 71.29 ± 9.78 | 0.302 |

TS, turner syndrome; HVs, healthy age-matched volunteer controls; BMI, body mass index; BSA, body surface area; IVSd, interventricular septum thickness at end-diastole; IVSs, interventricular septum thickness at end-systole; LVIDd, left ventricular internal dimension at end-diastole; LVIDs, left ventricular internal dimension at end-systole; LVPWd, left ventricular posterior wall thickness at end-diastole; LVPWs, left ventricular posterior wall thickness at end-systole; EDV, left ventricular end-diastolic volume; ESV, left ventricular end-systolic volume; SV, stroke volume; EF, ejection fraction; FS, fractional shortening, LV Mass, left ventricular end-diastolic mass; mean ± standard deviation is used for normally distributed variables and median (minimum/maximum) for non-normally distributed variables

* p-value <0.05

** p-value <0.001

and BSA (m$^2$) (P = 0.015). However, no differences were found between the two groups, with regard to mean age in Z-scores of cardiac structures, Z-scores of the aorta, EDV (BSA) (ml/m$^2$), ESV (BSA) (ml/m$^2$), SV (BSA) (ml/m$^2$), EF (%), FS (%), LV Mass (BSA) (g/m$^2$), MAPSE (mm), SBP (mmHg), and DBP (mmHg). Karyotype and cardiovascular morbidity of the TS group are summarized in **Table 2**. TS patients with CHD displayed when compared to non-CHD TS patients, a significantly higher sinotubular junction Z-score. Detailed results on clinical characteristics of CHD and non-CHD TS patients are presented in **Table 3**.

## MicroRNA microarray profiling between TS patients and HVs

Using the high-throughput SurePrint G3 Human v21 miRNA microarray platform, 60 miRNAs were found to be differentially abundant (P < 0.05, FDR adjusted) in the blood of TS patients compared to HVs. As shown in **Table 4**, of the 60 differentially abundant miRNAs, 41

**Table 2. Karyotype and cardiovascular morbidity in patients with TS.**

| Parameter | Turner Syndrome (n = 33) |
|---|---|
| Karyotype | |
| 45. X0 (%) | 17 (51.5) |
| Mosaic Form (%) | 11 (33.3) |
| Structural Chromosomal Aberration (%) | 2 (6.1) |
| Unspecified (Q.96.9) (%) | 3 (9.1) |
| CHD (%) | 13 (39.4) |
| AVD (%) | 11 (33.3) |
| BAV (%) | 10 (30.3) |
| MAV (%) | 1 (3.0) |
| CoA (%) | 5 (15.2) |
| Aortic Dilatation (%) | 1 (3.0) |
| PAPVC (%) | 4 (12.1) |
| Heart Operation (%) | 5 (15.2) |
| Weight Classification | |
| Normal Weight (%) | 18 (54.5) |
| Overweight (%) | 9 (27.3) |
| Obese (%) | 6 (18.2) |
| Arterial Hypertension (%) | 5 (15.2) |
| Carbohydrate Metabolism Disorder (%) | 3 (9.1) |

TS, turner syndrome; CHD, congenital heart disease; AVD, congenital aortic valve disease; BAV, bicuspid aortic valve; MAV, monocuspid aortic valve; CoA, coarctation of the aorta; PAPVC, partial anomalous pulmonary venous connection

miRNAs were significantly higher in patients with TS *versus* HVs, whereas the abundance levels of 19 miRNAs were significantly lower in patients with TS *versus* HVs. The majority of differentially abundant miRNAs (48 of 60 miRNAs) fell into the fold change range of 1.50–1.99 fold lower- or higher-abundance level). Besides, 12 miRNAs including 9 miRNAs (miR-374b-5p, miR-199a-5p, miR-340-3p, miR-125b-5p, miR-30e-3p, miR-126-3p, miR-99b-5p, miR-5695, and miR-26b-5p) with higher abundance level, and another 3 miRNAs (miR-6085, miR-5739, and miR-3656) with lower abundance level displayed an abundance level with fold changes ≥2.0-fold (**Table 4**). Hierarchical cluster analysis with the Euclidian distance could not discriminate accurately between TS patients and HVs. Specifically, as illustrated in **S1 Fig**, a group of miRNAs were found abundant in the TS patients group only and/or found abundant at a low level in HVs and vice versa. More detailed discrimination between TS patients and HVs based on the clustering dendrogram was, however, not possible.

## Validation of selected miRNAs abundance level in TS patients and HVs by qRT-PCR

We verified by RT-qPCR the abundance levels of the twelve selected miRNAs in all blood samples of TS patients (n = 33) and HVs (n = 14). The RT-qPCR of the validation experiments showed results that were largely concordant with the screening assays both in terms of adjusted *P*-value and of the direction of regulation i.e. up- or down-regulation. As shown in **Fig 1,** eight miRNAs, miR-374b-5p, miR-199a-5p, miR-340-3p, miR-125b-5p, miR-30e-3p, miR-126-3p, miR-5695, and miR-26b-5p showed a statistically significant higher abundance level, however,

**Table 3. Clinical characteristics of TS patients with and without CHD.**

| Parameter | CHD (n = 13) | Without CHD (n = 20) | P value |
|---|---|---|---|
| Age [years] | 20.43 ± 10.75 | 16.44 ± 4.77 | 0.227 |
| Height [cm] | 145.15 ± 13.41 | 149.05 ± 10.32 | 0.354 |
| Weight [kg] | 49.75 ± 16.68 | 55.19 ± 18.73 | 0.403 |
| BMI [kg/m$^2$] | 22.97 ± 4.86 | 24.50 ± 6.73 | 0.486 |
| BSA (m$^2$) | 1.40 ± 0.30 | 1.50 ± 0.28 | 0.374 |
| Z-Scores of Cardiac Structures | | | |
| IVSd | 0.95 ± 0.87 | 0.65 ± 0.94 | 0.360 |
| IVSs | 0.61 ± 0.69 | 0.38 ± 0.94 | 0.447 |
| LVIDd | -0.46 ± 0.50 | -0.81 ± 1.10 | 0.233 |
| LVIDs | -0.38 ± 0.74 | -0.46 ± 1.05 | 0.809 |
| LVPWd | 0.89 ± 0.74 | 0.42 ± 1.19 | 0.207 |
| LVPWs | -0.40 ± 1.08 | -0.56 ± 0.88 | 0.659 |
| Z-Scores of the Aorta | | | |
| Aortic Valve | 0.96 ± 1.90 | -0.08 ± 0.95 | 0.098 |
| Sinuses | 1.11 ± 1.67 | 0.51 ± 0.96 | 0.271 |
| Sinotubular Junction | 1.87 ± 1.57 | 0.61 ± 1.22 | **0.016**[*] |
| Ascending Aorta | 2.17 ± 2.17 | 0.72 ± 1.29 | **0.052** |
| EDV (BSA) [ml/m$^2$] | 55.93 ± 11.60 | 54.81 ± 12.78 | 0.801 |
| ESV (BSA) [ml/m$^2$] | 18.49 ± 5.38 | 18.59 ± 5.88 | 0.963 |
| SV (BSA) [ml/m$^2$] | 39.56 ± 6.85 | 36.20 ± 8.73 | 0.250 |
| EF [%] | 68.23 ± 7.10 | 66.20 ± 6.09 | 0.387 |
| FS [%] | 38.00 (27.00/47.00) | 35.00 (30.00/49.00) | 0.396 |
| LV Mass (BSA) [g/m$^2$] | 76.00 ± 17.67 | 67.93 ± 11.38 | 0.126 |
| MAPSE [mm] | 15.08 ± 2.87 | 14.90 ± 2.53 | 0.854 |
| SBP [mmHg] | 124.15 ± 16.47 | 121.55 ± 13.39 | 0.622 |
| DBP [mmHg] | 77.54 ± 13.66 | 73.75 ± 12.02 | 0.408 |

CHD, congenital heart diseases; BMI, body mass index; BSA, body surface area; IVSd, interventricular septum thickness at end-diastole; IVSs, interventricular septum thickness at end-systole; LVIDd, left ventricular internal dimension at end-diastole; LVIDs, left ventricular internal dimension at end-systole; LVPWd, left ventricular posterior wall thickness at end-diastole; LVPWs, left ventricular posterior wall thickness at end-systole; EDV, left ventricular end-diastolic volume; ESV, left ventricular end-systolic volume; SV, stroke volume; EF, ejection fraction; FS, fractional shortening, LV Mass, left ventricular end-diastolic mass, MAPSE, mitral annular plane systolic excursion, SBP, systolic blood pressure, DBP, diastolic blood pressure. Mean ± standard deviation is used for normally distributed variables and median (minimum/maximum) for non-normally distributed variables

[*] p-value <0.05

there were no significant differences in abundance level for miR-99b-5p, miR-6085, miR-5739, and miR-3656.

## Correlation of miRNAs with clinical data

To study whether there were correlations between the dysregulated and validated miRNAs by RT-qPCR and the different clinical characteristics, correlation analyses between the miRNA levels and clinical data were tested. The results showed that the abundance levels of miR-374b-5p, miR-199a-5p, miR-125b-5p, miR-30e-3p and miR-126-3p were correlated with different parameters as shown in Table 5. Specifically, miR-125b-5p was correlated with IVSd (r = -0.39; p = 0.0243), miR-199a-5p with LVIDd (r = 0.38; p = 0.0352), miR-126-3p with sinotubular junction Z-score (r = 0.42; p = 0.0154), EF (r = 0.37; p = 0.0348), FS (r = 0.37; p = 0.0356) and MAPSE (r = 0.42; p = 0.0150), and miR-374b-5p with EDV (BSA) (r = 0.40; p = 0.0303)

**Table 4. Significantly abundant miRNAs in the blood of patients with TS (n = 33) compared to HVs (n = 14) as determined by microarray.**

| miRNA | Median TS | Median HVs | Log Difference | Fold Change | Regulation | P-value | Corrected P-value | AUC |
|---|---|---|---|---|---|---|---|---|
| miR-374b-5p | 4.24 | 2.52 | 1.72 | 3.29 | Up | 0.00256 | 0.04182 | 0.74 |
| miR-199a-5p | 6.33 | 5.01 | 1.32 | 2.50 | Up | 0.00017 | 0.01606 | 0.80 |
| miR-340-3p | 5.31 | 3.99 | 1.32 | 2.50 | Up | 0.00061 | 0.02037 | 0.78 |
| miR-125b-5p | 6.33 | 5.02 | 1.30 | 2.47 | Up | 0.00081 | 0.02425 | 0.78 |
| miR-30e-3p | 4.68 | 3.41 | 1.27 | 2.41 | Up | 0.00013 | 0.01456 | 0.81 |
| miR-126-3p | 5.36 | 4.11 | 1.25 | 2.38 | Up | 0.00231 | 0.04042 | 0.75 |
| miR-99b-5p | 3.82 | 2.69 | 1.13 | 2.20 | Up | 0.00073 | 0.02251 | 0.76 |
| miR-5695 | 2.00 | 0.92 | 1.08 | 2.11 | Up | $3.00E^{-06}$ | 0.00383 | 0.85 |
| miR-26b-5p | 6.33 | 5.32 | 1.00 | 2.01 | Up | $9.60E^{-05}$ | 0.01288 | 0.82 |
| miR-215-5p | 8.16 | 7.20 | 0.96 | 1.95 | Up | 0.00040 | 0.01804 | 0.79 |
| miR-505-3p | 4.56 | 3.61 | 0.96 | 1.94 | Up | $2.14E^{-06}$ | 0.00383 | 0.86 |
| miR-378f | 2.05 | 1.10 | 0.95 | 1.93 | Up | 0.00039 | 0.01804 | 0.81 |
| miR-148b-3p | 5.02 | 4.09 | 0.94 | 1.91 | Up | 0.00037 | 0.01741 | 0.76 |
| miR-454-3p | 3.42 | 2.51 | 0.92 | 1.89 | Up | 0.00030 | 0.01678 | 0.83 |
| miR-99a-5p | 3.49 | 2.59 | 0.90 | 1.87 | Up | 0.00326 | 0.04646 | 0.77 |
| miR-378a-5p | 5.44 | 4.55 | 0.89 | 1.85 | Up | 0.00011 | 0.01288 | 0.81 |
| miR-193a-5p | 2.17 | 1.30 | 0.87 | 1.83 | Up | $8.53E^{-05}$ | 0.01288 | 0.81 |
| miR-155-5p | 3.64 | 2.77 | 0.87 | 1.83 | Up | 0.00017 | 0.01606 | 0.80 |
| miR-194-5p | 7.99 | 7.12 | 0.87 | 1.82 | Up | 0.00175 | 0.03614 | 0.77 |
| miR-146b-5p | 2.49 | 1.64 | 0.85 | 1.80 | Up | 0.00100 | 0.02719 | 0.85 |
| miR-28-5p | 4.63 | 3.78 | 0.84 | 1.79 | Up | 0.00050 | 0.02017 | 0.80 |
| miR-550b-2-5p | 2.59 | 1.75 | 0.84 | 1.79 | Up | 0.00058 | 0.02017 | 0.76 |
| miR-133b | 2.99 | 2.17 | 0.83 | 1.77 | Up | 0.00054 | 0.02017 | 0.81 |
| miR-4659b-3p | 1.87 | 1.06 | 0.80 | 1.75 | Up | 0.00343 | 0.04681 | 0.79 |
| miR-361-5p | 6.29 | 5.49 | 0.80 | 1.75 | Up | 0.00025 | 0.01671 | 0.85 |
| miR-181b-5p | 3.68 | 2.90 | 0.78 | 1.72 | Up | 0.00011 | 0.01288 | 0.83 |
| miR-195-5p | 3.08 | 2.30 | 0.77 | 1.71 | Up | $9.73E^{-05}$ | 0.01288 | 0.84 |
| miR-23b-3p | 6.19 | 5.43 | 0.76 | 1.69 | Up | 0.00091 | 0.02566 | 0.75 |
| miR-942-5p | 4.35 | 3.60 | 0.75 | 1.68 | Up | $4.66E^{-05}$ | 0.01275 | 0.81 |
| miR-181a-2-3p | 2.53 | 1.82 | 0.72 | 1.64 | Up | 0.00085 | 0.02522 | 0.79 |
| miR-132-3p | 4.40 | 3.69 | 0.71 | 1.64 | Up | 0.00010 | 0.01288 | 0.81 |
| miR-17-3p | 3.87 | 3.17 | 0.70 | 1.63 | Up | 0.00029 | 0.01678 | 0.77 |
| miR-335-5p | 1.71 | 1.02 | 0.69 | 1.62 | Up | 0.00193 | 0.03782 | 0.80 |
| miR-128-3p | 6.55 | 5.87 | 0.68 | 1.60 | Up | 0.00337 | 0.04681 | 0.72 |
| miR-500b-5p | 3.36 | 2.71 | 0.65 | 1.57 | Up | 0.00018 | 0.01650 | 0.78 |
| miR-4659a-3p | 2.44 | 1.81 | 0.63 | 1.54 | Up | 0.00302 | 0.04562 | 0.75 |
| miR-186-5p | 7.20 | 6.58 | 0.62 | 1.54 | Up | 0.00058 | 0.02017 | 0.79 |
| miR-942-3p | 2.46 | 1.84 | 0.62 | 1.53 | Up | 0.00226 | 0.04042 | 0.72 |
| miR-1255b-5p | 2.24 | 1.63 | 0.60 | 1.52 | Up | 0.00020 | 0.01671 | 0.76 |
| miR-222-3p | 4.42 | 3.82 | 0.60 | 1.52 | Up | 0.00033 | 0.01678 | 0.73 |
| miR-30a-5p | 5.41 | 4.82 | 0.59 | 1.51 | Up | 0.00319 | 0.04621 | 0.73 |
| miR-6085 | 4.88 | 5.96 | -1.08 | 2.11 | Down | 0.00149 | 0.03335 | 0.75 |
| miR-5739 | 5.36 | 6.40 | -1.04 | 2.05 | Down | 0.00103 | 0.02720 | 0.75 |
| miR-3656 | 4.38 | 5.40 | -1.02 | 2.03 | Down | $7.12E^{-05}$ | 0.01288 | 0.82 |
| miR-8069 | 10.52 | 11.49 | -0.97 | 1.95 | Down | $7.53E^{-06}$ | 0.00508 | 0.87 |
| miR-2861 | 3.85 | 4.74 | -0.89 | 1.86 | Down | 0.00022 | 0.01671 | 0.80 |
| miR-6789-5p | 2.48 | 3.35 | -0.87 | 1.83 | Down | 0.00044 | 0.01866 | 0.75 |

*(Continued)*

**Table 4.** (Continued)

| miRNA | Median TS | Median HVs | Log Difference | Fold Change | Regulation | P-value | Corrected P-value | AUC |
|---|---|---|---|---|---|---|---|---|
| miR-6803-5p | 5.89 | 6.75 | -0.86 | 1.82 | Down | 0.00024 | 0.01671 | 0.80 |
| miR-6127 | 8.07 | 8.93 | -0.85 | 1.81 | Down | 0.00054 | 0.02017 | 0.81 |
| miR-1275 | 2.51 | 3.32 | -0.81 | 1.75 | Down | 0.00105 | 0.02735 | 0.79 |
| miR-6749-5p | 4.78 | 5.58 | -0.81 | 1.75 | Down | 0.00034 | 0.01678 | 0.78 |
| miR-6089 | 6.69 | 7.50 | -0.80 | 1.74 | Down | $3.50E^{-05}$ | 0.01275 | 0.84 |
| miR-6869-5p | 4.22 | 4.98 | -0.76 | 1.69 | Down | $5.00E^{-05}$ | 0.01275 | 0.80 |
| miR-3960 | 6.69 | 7.44 | -0.74 | 1.68 | Down | $9.97E^{-06}$ | 0.00508 | 0.89 |
| miR-6165 | 4.12 | 4.84 | -0.72 | 1.65 | Down | 0.00064 | 0.02037 | 0.76 |
| miR-6087 | 5.63 | 6.31 | -0.68 | 1.61 | Down | $9.73E^{-06}$ | 0.00508 | 0.85 |
| miR-3162-5p | 4.63 | 5.30 | -0.67 | 1.60 | Down | 0.00033 | 0.01678 | 0.79 |
| miR-4459 | 5.67 | 6.34 | -0.67 | 1.59 | Down | 0.00015 | 0.01606 | 0.81 |
| miR-4507 | 4.93 | 5.53 | -0.61 | 1.52 | Down | 0.00048 | 0.02006 | 0.78 |
| miR-210-3p | 7.48 | 8.07 | -0.60 | 1.51 | Down | 0.00180 | 0.03630 | 0.76 |

TS, turner syndrome; HVs, healthy age-matched volunteer controls; AUC area under the receiver operating characteristic curve, unpaired two-tailed t test, >1.5-fold difference and Benjamini-Hochberg FDR $P \leq 0.05$

and MAPSE (r = 0.39; p = 0.0368). However, no significant correlations were found between abundance levels of miR-340-3p, miR-5695, and miR-26b-5p and clinical parameters.

No significant differences were observed between the abundance levels of validated miR-NAs i.e. miR-374b-5p, miR-199a-5p, miR-340-3p, miR-125b-5p, miR-30e-3p, and miR-26b-5p and coarctation of the aorta (CoA), monocuspid valve, dilatation of the aorta, partial anomalous pulmonary venous connection (PAPVC), and disorders of carbohydrate metabolism. However, a significant difference was observed only between abundance levels of miR-5695 and miR-126-3p in TS patients with and without CHD and BAV, respectively. Specifically, a significantly higher abundance level of miR-5695 was observed in TS patients with CHD compared to TS patients without CHD (adjusted p = 0.0265; log2-fold change 1.99) (Fig 2), whereas a significantly lower abundance level of miR-126-3p in TS patients with AVD compared to patients without AVD (p = 0.0139, log2-fold change 1.52) (Fig 2).

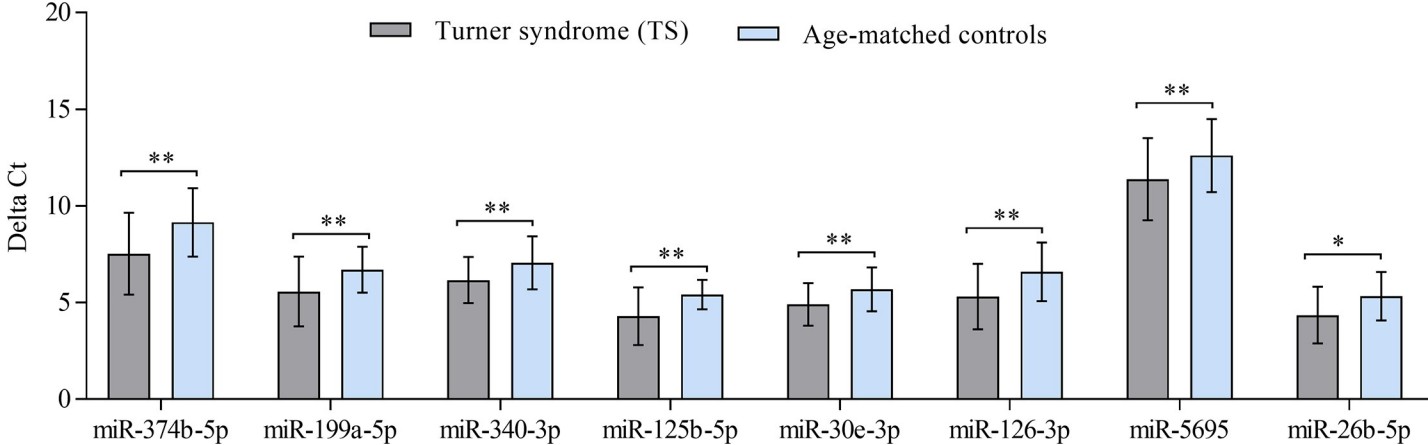

**Fig 1. Validation of eight differentially expressed miRNAs in the blood of patients with TS (n = 33) compared to HVs (n = 14) as determined by RT-qPCR (*P* < 0.05).** Mean ΔCt (Lower ΔCt, higher abundance level). RNAU6B as an endogenous control for normalization, Unpaired-two-tailed t-tests and median ± standard deviation (STDV) were used to evaluate differences in abundance. * P ≤ 0.05; ** P ≤ 0.01; *** P ≤ 0.001.

**Table 5. Correlation of validated miRNAs by RT-qPCR with clinical parameters.**

| Parameters | miR-374b-5p | | miR-199a-5p | | miR-125b-5p | | miR-30e-3p | | miR-126-3p | |
|---|---|---|---|---|---|---|---|---|---|---|
| | r | p-value | r | p-value | r | p-value | r | p-value | r | p-value |
| Z-Scores of Cardiac Structures | - | ns | - | ns | - | ns | - | ns | - | ns |
| IVSd | - | ns | - | ns | -0.39 | 0.0243 | - | ns | - | ns |
| IVSs | - | ns | - | ns | - | ns | - | ns | - | ns |
| LVIDd | - | ns | 0.38 | 0.0352 | - | ns | - | ns | - | ns |
| LVIDs | - | ns | - | ns | - | ns | - | ns | - | ns |
| LVPWd | - | ns | - | ns | - | ns | - | ns | - | ns |
| LVPWs | - | ns | - | ns | - | ns | - | ns | - | ns |
| Z-Scores of the Aorta | - | ns | - | ns | - | ns | - | ns | - | ns |
| Aortic Valve | - | ns | - | ns | - | ns | - | ns | - | ns |
| Sinuses | - | ns | - | ns | - | ns | - | ns | - | ns |
| Sinotubular Junction | - | ns | - | ns | - | ns | - | ns | 0.42 | 0.0154 |
| Ascending Aorta | - | ns | - | ns | - | ns | - | ns | - | ns |
| EDV (BSA) [ml/m$^2$] | 0.40 | 0.0303 | - | ns | - | ns | - | ns | - | ns |
| ESV (BSA) [ml/m$^2$] | - | ns | - | ns | - | ns | - | ns | - | ns |
| SV (BSA) [ml/m$^2$] | - | ns | - | ns | - | ns | - | ns | - | ns |
| EF [%] | - | ns | - | ns | - | ns | - | ns | 0.37 | 0.0348 |
| FS [%] | - | ns | - | ns | - | ns | - | ns | 0.37 | 0.0356 |
| LV Mass (BSA) [g/m$^2$] | - | ns | - | ns | - | ns | - | ns | - | ns |
| MAPSE [mm] | 0.39 | 0.0368 | - | ns | - | ns | - | ns | 0.42 | 0.0150 |
| SBP [mmHg] | - | ns | - | ns | - | ns | - | ns | - | ns |
| DBP [mmHg] | 0.37 | 0.0498 | - | ns | - | ns | 0.37 | 0.0362 | 0.37 | 0.0364 |

IVSd, interventricular septum thickness at end-diastole; IVSs, interventricular septum thickness at end-systole; LVIDd, left ventricular internal dimension at end-diastole; LVIDs, left ventricular internal dimension at end-systole; LVPWd, left ventricular posterior wall thickness at end-diastole; LVPWs, left ventricular posterior wall thickness at end-systole; EDV, left ventricular end-diastolic volume; ESV, left ventricular end-systolic volume; SV, stroke volume; EF, ejection fraction; FS, fractional shortening, LV Mass, left ventricular end-diastolic mass, MAPSE, mitral annular plane systolic excursion, SBP, systolic blood pressure, DBP, diastolic blood pressure, ns, non-significant, p-value <0.05.

## Comparative pathway analysis

MiRTargetLink indicated 9 genes with "strong" evidence being targets for the miR-199a-5p, miR-125b-5p, miR-30e-3p, miR-126-3p, and miR-26b-5p as shown in **Fig 3**. A strong interaction was observed between miR-199a-5p with 7 genes (*EZH2*, *SMAD4*, *ERBB2*, *SIRT1*, *ERBB3*, *PTGS2*, and *VEGFA*), miR-125b-5p with 4 genes (*ERBB3*, *ERBB2*, *BCL2*, and *SMAD4*), miR-30e-3p with 1 gene (*NFKBIA*), miR-126-3p with 4 genes (*VEGFA*, *BCL2*, *NFKBIA*, and *SIRT1*), and miR-26b-5p with 2 genes (*PTGS2* and *EZH2*), which are listed in the resulting network **Fig 3**.

## Discussion

In this study, 41 miRNAs with higher abundance and 19 miRNAs with lower abundance were found in TS patients compared to HVs. Eight miRNAs (miR-374b-5p, miR-199a-5p, miR-340-3p, miR-125b-5p, miR-30e-3p, miR-126-3p, miR-5695, and miR-26b-5p) exhibited more than 2-fold changes in their abundance level and were validated by RT-qPCR. Since cardiovascular morbidities and their long term complications are the main cause of early death in TS patients, the establishment of predictive cardiovascular markers might be beneficial for risk stratification and therapeutic intervention in TS [28]. However, prospective long-term studies with

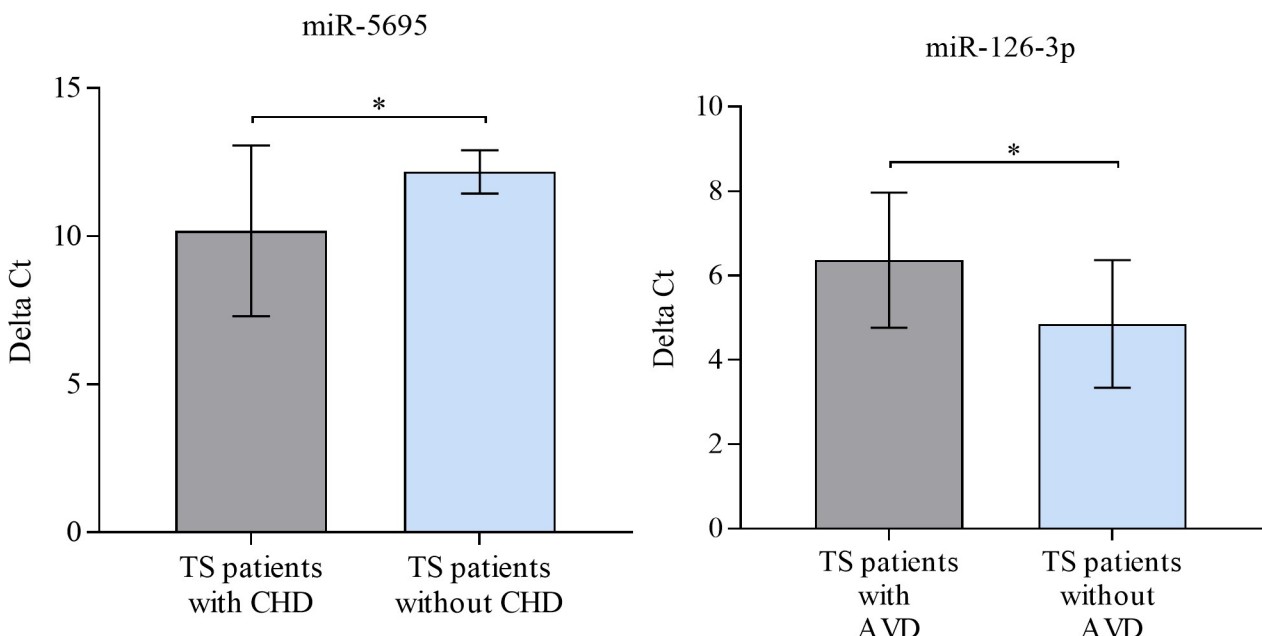

**Fig 2. Differentially expressed miRNA in TS patients with (n = 12) and without CHD (n = 21), and patients with (n = 10) and without AVD (n = 23).** Mean ΔCt (Lower ΔCt, higher abundance level). RNAU6B as an endogenous control for normalization, Unpaired-two-tailed t-tests and median ± standard deviation (STDV) were used to evaluate differences in abundance. * $P \leq 0.05$; ** $P \leq 0.01$; *** $P \leq 0.001$.

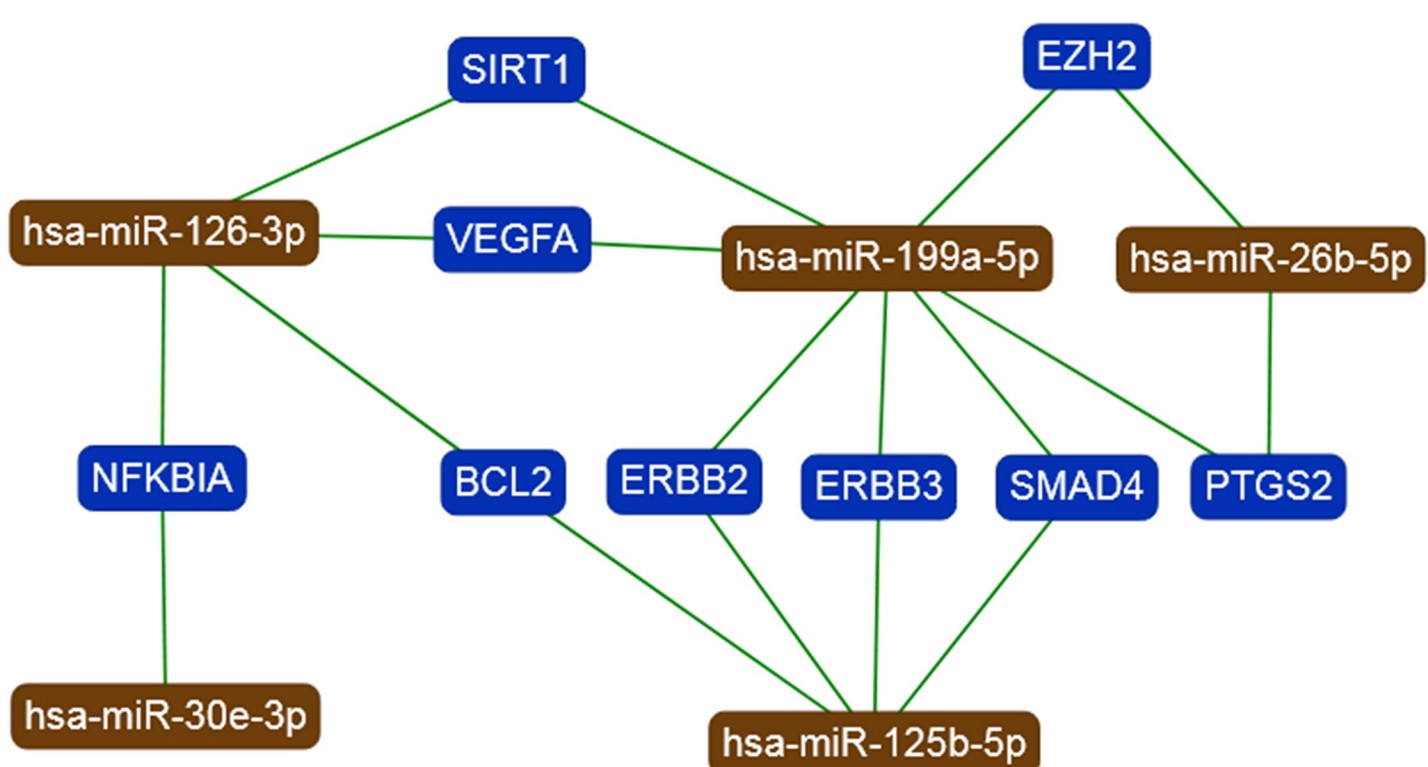

**Fig 3. Target network for the five validated miRNAs in brown, validated target genes are presented in blue.**

larger cohorts are required to gain further insights into the specific functions of these miRNAs in TS patients with and without CHD and their possible impact on CHD-associated cardiovascular complications in later life. Of the dysregulated validated miRNAs, miR-5695 was significantly higher in TS patients with CHD as compared to TS patients without CHD whereas, miR-126-3p was significantly lower in TS patients with AVD compared to TS patients without AVD. MiR-5695 has not yet been observed to be involved in any biological function in the CHD and/or cardiovascular manifestations. Additional experimental and clinical studies are necessary to confirm whether this newly found miRNA is specific for TS-associated CHD.

MiR-126 is an endothelial cell-restricted miRNA, which mediates developmental angiogenesis *in vivo* [29], essential for vascular endothelium and endocardium cell signaling and promotes migration, proliferation, and network vessel formation *in vitro* [30]. Higher circulating level of miR-126 serves as biomarkers for vascular damage [31], and the circulating level of miR-126 was considered as a biomarker in patients with acute myocardial infarction [32]. In Marfan syndrome (MFS) patients, structural alterations of the aorta and other arterial vascular system disorders determine the mid- and long-term morbidity and mortality. In these patients, we found that the abundance level of miR-126-3p was significantly higher compared to healthy controls [24], suggesting that the higher abundance level of miR-126-3p may be related to vascular morbidities, including vessel wall disease, leading to the well-known cardiovascular complications, including aortic aneurysm and dissection [24, 33]. In MFS patients, increased elasticity and dilatation of the aorta leads to aneurysms and dissections in large arterial vessels. In TS patients, however, elevated stiffness of the main arterial vessels [34] might lead to higher blood pressure and increase risk of coronary heart disease [35]. The precise molecular mechanisms and the role of the detected miR-126-3p have to be evaluated in further clinical and experimental settings. Nevertheless, miR-126-3p was involved in the development of thoracic aortic aneurysm in adults [36]. MiR-126-3p was found reduced in pregnant women with preeclampsia, where endothelial vascular injury in different organs occurs, resulting in higher blood pressure and renal vascular injury and proteinuria [37]. Thus, the downregulation of mR-126-3p in our TS patients with cardiovascular morbidities, presenting usually with higher blood pressure, increased risk of coronary heart disease and thoracic aortic dissection, may indicate structural and functional endothelial alterations and vessel disease in TS patients.

Presumably, miR-126-3p is involved in the embryological process of the aortic valve and aortic development due to a statistically significant downregulation among TS patients with AVD compared to TS patients without AVD as observed in the present study. The positive correlation between the downregulated miR-126-3p and the sinotubular junction Z-score within the TS cohort may be due to possibly altered aortic developments. Dilatation of the ascending aorta is more common in patients with BAV [38]. In other patients without TS and BAV, this phenomenon might be due to hemodynamic flow abnormalities associated with bicuspid aortic valve [39].

Many target genes implicated in heart development and structural heart malformations have been identified as experimentally validated targets of the differentially expressed miRNAs. Of these targets, *PTGS2* (prostaglandin-endoperoxide synthase 2) is required for normal cardiac development and its related pathways [40]. The Bcl-2 family is a key regulator in cell apoptosis, among which the *Bcl-2* gene has an anti-apoptotic effect by regulating the cytochrome-c in the activation of the apoptotic intrinsic pathway [41]. *Bcl-2* has been validated to target 2 miRNAs in our study, including miR-126-3p and miR-125b-5p. These two miRNAs suppress apoptosis in certain cells by targeting and regulating the expression level of *Bcl-2* [42–44]. Thus, the expression level of miR-126-3p and miR-125b-5p in the heart and vascular endothelial may have a similar function in influencing the apoptotic pathway. In addition, the product of genes encoding tissue inhibitors of matrix metalloproteinases (TIMPs) [*TIMP1*

(Metallopeptidase Inhibitor 1) and *TIMP3* (Metallopeptidase Inhibitor 3)] were identified as risk genes for BAV and aortopathy in patients with TS [45]. These genes along the *MMP* (matrix metalloproteinases) gene play an important role in the development of the aortic valve and protect aortic tissue integrity [45]. An imbalance in the TIMP/MMP ratio in patients with TS increases the risk for both congenital cardiovascular defects and later onset aortic disease [45].

## Study limitations

Limitations of our study are related to a limited sample size. The reported miRNAs need to be further validated in a larger cohort of TS patients with different ages including neonates and infants to confirm the abundance level of the identified miRNAs. In addition, further validation studies on the mRNA and protein levels are needed to find out the relationship between both miRNAs and their target genes in TS patients with CHD.

## Conclusion

In this study, we have identified circulating miRNAs that serve as a molecular signature in patients with TS with and without CHD compared to HVs. These differentially abundant miRNAs showed a significant correlation to the clinical parameters of these patients. These findings may lead to more profound insights into the development of cardiovascular morbidities which are associated with TS and may guide the development of novel diagnostic approaches and preventive strategies.

## Supporting information

**S1 Fig. Unsupervised hierarchical clustering (Euclidian distance, complete linkage) of the patients with TS compared to HVs based on the abundance of the 50 with the highest variance.**
(TIF)

## Author Contributions

**Conceptualization:** Masood Abu-Halima, Eckart Meese, Hashim Abdul-Khaliq.

**Data curation:** Masood Abu-Halima, Felix Sebastian Oberhoffer, Mohammed Abd El Rahman, Anna-Maria Jung, Michael Zemlin, Tilman R. Rohrer, Eckart Meese.

**Formal analysis:** Masood Abu-Halima, Mustafa Kahraman, Andreas Keller.

**Funding acquisition:** Masood Abu-Halima, Eckart Meese, Hashim Abdul-Khaliq.

**Investigation:** Masood Abu-Halima.

**Methodology:** Masood Abu-Halima, Felix Sebastian Oberhoffer, Mohammed Abd El Rahman, Anna-Maria Jung, Michael Zemlin, Tilman R. Rohrer.

**Project administration:** Masood Abu-Halima, Hashim Abdul-Khaliq.

**Resources:** Masood Abu-Halima, Mohammed Abd El Rahman, Eckart Meese.

**Software:** Masood Abu-Halima, Mustafa Kahraman, Andreas Keller.

**Supervision:** Masood Abu-Halima, Eckart Meese, Hashim Abdul-Khaliq.

**Validation:** Masood Abu-Halima.

**Visualization:** Masood Abu-Halima, Michael Zemlin, Tilman R. Rohrer, Eckart Meese, Hashim Abdul-Khaliq.

**Writing – original draft:** Masood Abu-Halima, Felix Sebastian Oberhoffer, Eckart Meese, Hashim Abdul-Khaliq.

**Writing – review & editing:** Masood Abu-Halima, Felix Sebastian Oberhoffer, Eckart Meese, Hashim Abdul-Khaliq.

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
