## [Decision Letter · Decision Letter 0]

17 Mar 2020

PONE-D-20-01777

Insights from Circulating MicroRNAs in Cardiovascular Entities in Turner Syndrome Patients

PLOS ONE

Dear author,

Thank you for submitting your manuscript to PLOS ONE. After careful consideration, we feel that it has merit but does not fully meet PLOS ONE’s publication criteria as it currently stands. Therefore, we invite you to submit a revised version of the manuscript that addresses the points raised during the review process.

please follow closely instructions of the reviewers.

We would appreciate receiving your revised manuscript by 16 may 2020. To enhance the reproducibility of your results, we recommend that if applicable you deposit your laboratory protocols in protocols.io, where a protocol can be assigned its own identifier (DOI) such that it can be cited independently in the future. For instructions see: http://journals.plos.org/plosone/s/submission-guidelines#loc-laboratory-protocols

We look forward to receiving your revised manuscript.

Kind regards,

Laurent Metzinger

Academic Editor

PLOS ONE

Journal Requirements:

Reviewers' comments:

Reviewer's Responses to Questions

**Comments to the Author**

1. Is the manuscript technically sound, and do the data support the conclusions?

Reviewer #1: Yes

Reviewer #2: Yes

2. Has the statistical analysis been performed appropriately and rigorously? 

Reviewer #1: I Don't Know

Reviewer #2: Yes

3. Have the authors made all data underlying the findings in their manuscript fully available?

Reviewer #1: Yes

Reviewer #2: Yes

4. Is the manuscript presented in an intelligible fashion and written in standard English?

Reviewer #1: Yes

Reviewer #2: Yes

5. Review Comments to the Author

Reviewer #1: In this article ''Insights from circulating MicroRNAs in Cardiovascular entities in Turner Syndrome Patients'' by Abu-Halima et al. we find an interesting study about différences in microRNA expression in patients presenting Turner Syndrome (TS) associated to congenital heart disease. They first compared patients with TS to a matched cohort of healthy volunteers and identified 60 different microRNAs with either increased or decreased seric expression. After that, they compared the level of these microRNAs between patients with TS, presenting or not a congenital heart disease and found 8 different microRNAs between both groups, with miR-126-3p correlated to the sinotubular junction Z-score.

We appreciated reading this study and are favorable for publication.

Meanwhile, we have some comments and questions:

- Among microRNAs found to be different between TS and HVs, after the microArray profiling, how many are already described to be in relation with congenital heart disease? Maybe the 40 miRNAs found different are specific to other congenital abnormalities related to TS but not directly implicated in the latter diseases.

- These findings are interesting to elaborate a predictive preoperative markers of congenital heart disease in patients with TS. Meanwhile, are there any further plans to measure these miRNAs during pregnancy, in order to predict whether the TS embryo would present a conginetal disease or not, at birth?

- In table 4, there are 6 missing upregulated miR- in the first column. Please fill the gap in miR-column.

Reviewer #2: In this original study, the authors identified a profile of eight blood microRNAs capable of discriminating Turner Syndrome (TS) patients from healthy subjects. Some of these miRNAs are correlated with cardiac complications in TS. The main limitation of this study is the low number of samples, especially during the validation phase. It would have been interesting to carry out the validation step by qRT-PCR on a second cohort with a larger number of samples. Nonetheless, the study is innovative and well written. I only have the following minor comments :

- Page 2, line 20: delete "syndrome", the abbreviation TS is sufficient.

- Page 3, paragraph "Subjects": have the blood samples been centrifuged? If so, specify the speed, time and temperature of the centrifugation.

- Page 4, paragraph "Preparation of RNA and RNA quality control": describe briefly the RNA isolation, citing another publication is not enough. The pre-analytical phase represents a major source of variability when quantifying the expression of circulating miRNAs.

- Table 4: names of several miRNAs are missing.

6. PLOS authors have the option to publish the peer review history of their article (what does this mean?). If published, this will include your full peer review and any attached files.

Reviewer #1: No

Reviewer #2: Yes: Mustapha ZENDJABIL

---

## [Author Response · Author response to Decision Letter 0]

17 Mar 2020

Dear Co-Editors-in-Chief,

Dear Laurent Metzinger

We appreciate the comments of the reviewers and their efforts to further improve our manuscript. We revised the manuscript as suggested. Please find attached the revised manuscript and our point-by-point response. 

Sincerely,

Masood Abu-Halima

Reviewer #1:

In this article ''Insights from circulating MicroRNAs in Cardiovascular entities in Turner Syndrome Patients'' by Abu-Halima et al. we find an interesting study about differences in microRNA expression in patients presenting Turner Syndrome (TS) associated to congenital heart disease. They first compared patients with TS to a matched cohort of healthy volunteers and identified 60 different microRNAs with either increased or decreased expression. After that, they compared the level of these microRNAs between patients with TS, presenting or not a congenital heart disease and found 8 different microRNAs between both groups, with miR-126-3p correlated to the sinotubular junction Z-score. We appreciated reading this study and are favorable for publication.

Meanwhile, we have some comments and questions:

1. Among microRNAs found to be different between TS and HVs, after the microarray profiling, how many are already described to be in relation with congenital heart disease? Maybe the 60 miRNAs found different are specific to other congenital abnormalities related to TS but not directly implicated in the latter diseases. 

 As the reviewer correctly mentioned, one can only state that the found alterations in the miRNA profile may play an essential role in the pathophysiology of CHDs. However, it is remarkable, that our identified miRNAs have been reported to play a role in several cardiac pathologies, including CHDs. Nevertheless, the reviewer is of course right in that we cannot claim an essential role of these miRNAs in the pathophysiology of CHD. 

 In addition, out of 60 identified miRNAs, 24 miRNAs play a role in different CHD like miR-199a, miR-340, miR-125b, miR-126, miR-99b, miR-148b, miR-454, miR-99a, miR-155, miR-194, miR-28, miR-550b-2, miR-361, miR-181b, miR-23b, miR-942, miR-132, miR-17, miR-186, miR-222, miR-6085, miR-6789, miR-1275, and miR-210. These miRNAs have been identified by us and others, in monozygotic twins discordant for CHDs (PMID: 31805172), in patients with univentricular hearts (PMID: 31600281), in patients with repaired Tetralogy of Fallot with and without heart failure (PMID: 28693530), and in Marfan syndrome patients with cardiovascular manifestations (PMID: 28679133).

2. These findings are interesting to elaborate a predictive preoperative markers of congenital heart disease in patients with TS. Meanwhile, are there any further plans to measure these miRNAs during pregnancy, in order to predict whether the TS embryo would present a congenital disease or not, at birth?

 Although this is, of course, a final aim, it is, however, premature to draw conclusions about the use of our identified miRNAs to predict whether the TS embryo would present a CHD or not at birth. These are preliminary results to describe the expression patterns of miRNA in TS patients, with and with CHDs. 

 This study will be followed by a larger multicenter study including subgroups of CHD patients to confirm the expression level of the identified miRNAs, their role in CHD, and the possibility to use the miRNA as novel biomarkers to predict the presence or absence of CHD in TS patients. The description of the miRNA expression in this study is however justified. 

3. In table 4, there are 6 missing upregulated miRNAs in the first column. Please fill the gap in miR-column.

 We would like to apologize for this mistake. In the revised version, we now provide the missing miRNAs.

Reviewer #2: 

In this original study, the authors identified a profile of eight blood microRNAs capable of discriminating Turner Syndrome (TS) patients from healthy subjects. Some of these miRNAs are correlated with cardiac complications in TS. The main limitation of this study is the low number of samples, especially during the validation phase. It would have been interesting to carry out the validation step by qRT-PCR on a second cohort with a larger number of samples. 

 The reviewer is certainly right about the small sample size of the included subjects i.e. TS patients and age-matched controls, and that the screening and validation phase were carried out on the same sample. These are preliminary results to describe the expression pattern of miRNAs in TS patients with CHD. We are aware of such limitations. We would like to point out that the occurrence of CHD in TS is very rare and subsequently the number of included patients is relatively low. Nevertheless, significant differences in 60 expressed miRNA expression were found between the patients and matched controls. The description of the miRNA expression is however justified. Of course, we addressed this limitation in our revised manuscript.

Nonetheless, the study is innovative and well written. I only have the following minor comments:

1. Page 2, line 20: delete "syndrome", the abbreviation TS is sufficient.

 This has now been corrected in the manuscript.

2. Page 3, paragraph "Subjects": have the blood samples been centrifuged? If so, specify the speed, time and temperature of the centrifugation. 

 After collection of blood samples, tubes were kept for 24 hours at room temperature to lysis cells, followed by a −20°C storage for multiple days and then at −80°C for long-term storage. Centrifugation step has been done in the next step, in the RNA isolation. This has now been added to the ‘’Materials and Methods’’ section of the revised manuscript.

3. Page 4, paragraph "Preparation of RNA and RNA quality control": describe briefly the RNA isolation, citing another publication is not enough. The pre-analytical phase represents a major source of variability when quantifying the expression of circulating miRNAs.

 As requested by the reviewer, in the revised manuscript, we now provide a more concise paragraph about the ‘’Preparation of RNA and RNA quality control’’ with the changes highlighted.

4. Table 4: The names of several miRNAs are missing.

 We would like to apologize for this mistake. In the revised version, we now provide the missing miRNAs.

---

## [Editor Report · Decision Letter 1]

24 Mar 2020

Insights from Circulating MicroRNAs in Cardiovascular Entities in Turner Syndrome Patients

PONE-D-20-01777R1

Dear Author,

We are pleased to inform you that your manuscript has been judged scientifically suitable for publication and will be formally accepted for publication once it complies with all outstanding technical requirements.

With kind regards,

Laurent Metzinger

Academic Editor

PLOS ONE
---

## [Editor Report · Acceptance letter]

27 Mar 2020

PONE-D-20-01777R1 

Insights from Circulating MicroRNAs in Cardiovascular Entities in Turner Syndrome Patients 

Dear Dr. Abu-Halima:

I am pleased to inform you that your manuscript has been deemed suitable for publication in PLOS ONE. Congratulations! Your manuscript is now with our production department. 

With kind regards,

on behalf of

Professor Laurent Metzinger 

Academic Editor

PLOS ONE